# Current Guidelines for Protecting Health Workers from Occupational Tuberculosis Are Necessary, but Not Sufficient: Towards a Comprehensive Occupational Health Approach

**DOI:** 10.3390/ijerph17113957

**Published:** 2020-06-03

**Authors:** Rodney Ehrlich, Jerry M. Spiegel, Prince Adu, Annalee Yassi

**Affiliations:** 1Division of Occupational Medicine, School of Public Health and Family Medicine, University of Cape Town, Observatory, Cape Town 8001, South Africa; 2School of Population and Public Health, University of British Columbia, 2206 East Mall, Vancouver, BC V6T 1Z3, Canada; jerry.spiegel@ubc.ca (J.M.S.); prince.adu@alumni.ubc.ca (P.A.); annalee.yassi@ubc.ca (A.Y.)

**Keywords:** health workers, tuberculosis, occupational health, infection prevention and control, health system

## Abstract

Health workers globally are at elevated occupational risk of tuberculosis infection and disease. While a raft of guidelines have been published over the past 25 years on infection prevention and control (IPC) in healthcare, studies in different settings continue to show inadequate implementation and persistence of risk. The aim of this commentary is to argue, based on the literature and our own research, that a comprehensive occupational health approach is an essential complement to IPC guidelines. Such an approach includes a health system framework focusing on upstream or mediating components, such as a statutory regulation, leadership, an information system, and staff trained in protective disciplines. Within the classical prevention framework, primary prevention needs to be complemented by occupational health services (secondary prevention) and worker’s compensation (tertiary prevention). A worker-centric approach recognises the ethical implications of screening health workers, as well as the stigma perceived by those diagnosed with tuberculosis. It also provides for the voiced experience of health workers and their participation in decision-making. We argue that such a comprehensive approach will contribute to both the prevention of occupational tuberculosis and to the ability of a health system to withstand other crises of infectious hazards to its workforce.

## 1. What is the Problem?

High rates of tuberculosis (TB) in the populations of low- and middle-income countries (LMICs) are associated with high rates of latent TB infection (LTBI) and TB disease in health workers [1,2,3]. The most recent systematic review reports a pooled incidence rate ratio for active TB disease among health workers of 2.94, and a pooled odds ratio for LTBI of 2.27, relative to control populations [3]. In high-HIV-burden countries, HIV infection among health workers [4] sharply increases their risk of TB, while the rise in drug-resistant TB has further intensified the threat associated with the disease [5,6].

There is no shortage of prevention guidelines directed at healthcare settings where a TB hazard to staff and patients exists. International popularisation of the TB infection prevention and control (IPC) triad of administrative, environmental, and respiratory protection controls can be dated to the publication of guidelines in 1994 by the U.S. Centers for Disease Control and Prevention (CDC) [7]. The guidelines, updated in 2005, constituted a response to a resurgence of TB in the United States and nosocomial transmission in U.S. hospitals in the wake of the HIV epidemic [8,9,10]. In parallel, the World Health Organization (WHO) has published a series of guidelines for low-resource settings [11,12,13], as well as several supporting documents on the implementation of IPC [14,15] and provision of healthcare and related services for affected health workers [16]. 

IPC in healthcare settings has featured in other international responses. After considerable advocacy effort, including a statement by the International Commission for Occupational Health (ICOH) [17], the United Nations (UN) General Assembly Political Declaration of 2018 recognised healthcare workers as an occupational group at risk from TB, and called for IPC and TB screening and surveillance for this population [18]. 

While the core IPC guidelines were based on public health “first principles”, systematic reviews have concluded that evidence of the effectiveness of various protective measures is limited and/or of “low quality” [2,13,19]. The inability to provide data conforming to that produced, for example, in drug trials, arises from the practical and ethical difficulty of undertaking randomised controlled trials for prevention in this context. However, as argued in this commentary, successful preventive practices require an enabling system. An omnibus approach to the IPC package has emerged, reflecting the complex nature of such interventions [2,13,19]. Usefully, recent guidelines, have employed the Grading of Recommendations, Assessment, Development and Evaluation (GRADE) rating framework for public health and clinical recommendations which includes other sources of information and judgements—specifically, a balance of benefits and disadvantages, values and preferences, and resource requirements [13,20]. This enables “low quality” evidence on effect size to contribute to a “moderate” or “strong” recommendation if the other criteria are favourable. This approach thus incorporates contextual factors and uses an optimisation approach to applying evidence. 

However, the need for an approach that goes beyond IPC guidelines is suggested by the growing number of studies across the world that consistently reveal poor or inadequate implementation of TB IPC [21,22,23,24,25,26,27,28,29,30,31]. Barriers to IPC implementation vary with the study design and questions asked, but cover the whole gamut: lack of a national regulatory framework and associated budget; lack of management support; unfamiliarity of staff with IPC guidelines; failure to triage or screen patients; insufficient infrastructure and equipment, such as isolation spaces and personal protective equipment (PPE); deficient ventilation; inadequate staffing and training; poor functioning of infection control committees; and neglect of exposed non-clinical staff. Qualitative studies, which have a greater capacity to probe health workers’ experience, reveal a perception among health workers of a disproportionate focus on individual-level personal protections, particularly N95 respirators [24]; an experience of powerlessness [29]; habituation to TB risk or a sense of fatalism [26]; and difficulty in understanding patients and securing patient cooperation with IPC [21,28,31].

## 2. Objectives

The objectives of this piece are to argue for a comprehensive occupational health approach to the problem of TB in health workers, and to reflect on what such an approach adds to the prospects for improved prevention and practice. We draw on research carried out by our group as occupational health professionals and researchers in recent years [31,32,33,34,35,36,37,38,39,40,41,42], plus the experience of two of the authors (R.E. and A.Y.) as clinicians and policy advisors to healthcare facilities, as well as provincial and national public sector health departments in South Africa and Canada, respectively. 

We argue that a comprehensive approach should conceive of the problem of protecting health workers within the widest possible framework, including legal and ethical considerations, should be multilevel and cross-disciplinary, and should be informed by the experiences of health workers themselves. A schema for making this argument is given in Table 1. Each domain is dealt with in the sections that follow.

## 3. Levels of Disease Prevention

Figure 1 applies the framework of primary, secondary, and tertiary prevention to consider opportunities for protecting health workers from occupational TB. It directs attention to the number of ways in which health workers can be at risk and protected from TB, and is an antidote to exclusive focus on any one level. A strong health system is both an enabler and a consequence of prevention activities, as discussed further in the next section. 

Primary prevention in the form of IPC receives the lion’s share of attention in guidelines, and remains the foundation for protection. However, secondary prevention contributes to primary prevention by aiming to keep health workers unimpaired and non-infectious in their daily work. This includes occupational health services to provide screening for active TB and the management of affected health workers [16]. An occupational health platform would also be needed for any programme to screen for and treat LTBI. LTBI screening has long been recommended in low-TB-incidence countries, such as the United States [7,10], and appears in the latest WHO guideline as a (conditional) recommendation for low-TB-incidence countries only [43]. Consensus on what is required and feasible in high-TB-incidence, LMIC settings is elusive. While there are many studies of LTBI prevalence in these settings, programme implementation research in this area is scarce and is a pressing need.

Tertiary prevention, by ensuring that health workers with an occupational disease have access to medical care and special sick leave, and to rehabilitation or to supported retirement, should be regarded as a basic labour right and is discussed further in Section 5. 

Although not explicitly presented in Figure 1, HIV-infected health workers need to be considered within this framework as a particularly vulnerable sub-population needing protection. In South Africa, estimates of the proportion of health workers infected with HIV are of the order of 16% [4]. HIV infection dramatically increases the risk of progression from TB infection to disease [44]. A programme of voluntary HIV testing, followed by treatment and counselling on job placement, should thus be regarded as part of primary prevention of TB in affected settings.

Vaccination of health workers against TB also falls into the category of primary prevention. Bacille Calmette-Guerin (BCG) vaccination or revaccination of health workers is not currently a recommendation by either the CDC [10] or WHO, and variation in health worker BCG vaccination practice across Europe reflects the lack of consensus as to its efficacy [45]. However, vaccination should remain on the agenda of a comprehensive occupational health approach—for example, revaccination of LTBI-negative health workers or testing and rollout among health workers of one of the new vaccines on the horizon [46,47].

## 4. Health System Strengthening 

Using a health system framework involves a shift in perspective towards one that is cross-cutting and systemic, political as well as technical. It draws attention to the governance and organisation of healthcare that enable disease control and clinical programmes and practices, and which are geared to achieving greater equitability and sustainability in health outcomes. Health system assessment and strengthening are conceptualized by WHO as focused on the performance of six inter-related building blocks: governance and leadership, information, health financing, health workforce, services, and technology [48]. This is a two-way interaction. Adequate performance in respect of all the building blocks is required if IPC and workplace TB prevention programmes are to work, as discussed further below. Conversely, investment in protecting health workers from occupational infectious disease has the potential to yield system-wide benefits. These include TB surveillance as an indicator of respiratory disease risk in healthcare settings; “cross-silo” cooperation in the healthcare system; reduced absenteeism and improved staff retention and morale; and greater patient safety, quality of care, and trust in the health system [32,33,34,49,50]. 

At their most upstream, structural health system barriers encompass international and national political economy. An example particularly relevant to Africa is the imposition on governments of structural adjustment programmes by international lenders that require reducing the size of the public sector, thereby decreasing healthcare expenditures—including on the control of TB [51]. Zelnick [52] studied the struggles among South African nurses to provide care and protect themselves against bloodborne exposure in the early days of the HIV/AIDS epidemic. Among the causes of this situation, the author identifies the failure of the new South African government, under the pressure of neoliberal macroeconomic policies, to devote sufficient resources to district health facilities, resulting in staff shortages and a hazardous work environment over which nurses felt they had no control. More recently, Lispel and Fonn have illuminated the relatively unexplored subject of corruption in the health sector, particularly the diversion of health funding through rigged tendering and supply activities, combined with the enabling factor of poor governance [53].

A little more downstream, we recently used the WHO health systems framework from an occupational health perspective to explore the perceptions of key informants within the South African health system of barriers to protection of health workers from TB [38]. Such barriers include, inter alia, lack of an information system to produce the necessary intelligence on health and safety; fragmentation of governance across different organisational units within jurisdictions and health facilities; difficulty in maintaining technological components, such as germicidal ultraviolet light air disinfection; and lack of occupational health services trusted by staff.

Remedying the deficits outlined above involves costs in the form of organisational management, staffing, clinical practice, ancillary services, and procurement. Cost within fixed administrative budgets is, however, a major cross-cutting barrier [38,40]. This creates hesitancy among senior health service managers to commit significant resources to occupational health and safety amidst competing priorities, even if the actions are legally mandated, and especially if they believe TB in their staff is not an occupational disease [38,40]. This suggests that a useful starting point for collaboration between occupational health and IPC is an information system able to track and investigate active TB in health workers for the purposes of risk assessment and targeted IPC. An example is the Occupational Health and Safety Information System (OHASIS), developed through a collaboration of occupational health and IPC professionals at the University of British Columbia with the National Institute for Occupational Health in South Africa, and applied in the National Health Laboratory Service of South Africa [54,55]. 

## 5. Legal and Ethical Perspectives

Those applying generic guidelines need to consider the legal and related institutional environment of their jurisdictions, and whether this environment is sufficiently enabling. In many and perhaps most countries, occupational health and safety is governed by statutory regulation, which provides a framework of requirements and standards, as well as an enforcement mechanism to prevent occupational injuries and disease [56]. A review across Botswana, Zambia, and South Africa of laws relevant to reduction of TB transmission adopted a systems view by focusing on regulations governing national legal and policy frameworks; facility design, construction, and use; patients’ and health workers’ rights; and research, as well as the monitoring of infection control measures and TB surveillance among health workers. The authors concluded that the laws and regulations provided a “strong foundation” for these activities [57]. 

However, in high-TB-burden countries, particularly in public sector facilities, the competing demands on the healthcare budget may be used as an alibi for the failure to provide sufficient resources to protect staff from TB [38,40]. There may also be resistance to the application of an “industrial” model of regulation, inspection, and enforcement to the healthcare sector. The notion of voluntary acceptance of risk, an old common law defence by employers against liability for occupational injury or disease, and one that has long formed part of the vocational ethos of healthcare, contributes to this resistance [9,36]. Legislation by itself is thus no guarantee of adequate preventive practice in healthcare—as indicated, for example, by the number of studies on poor implementation of TB IPC in South Africa [23,24,28,29,31].

Worker’s compensation or broader social security are components of tertiary prevention of occupational TB. As with preventive legislation, practice varies. In South Africa, TB suffered by a health worker who has had contact with TB patients is presumed to be occupational. In contrast, in Mozambique, TB is not recognised by statute as an occupational disease, an omission identified by local health workers as a major barrier to comprehensive management of occupational TB [40]. Worker’s compensation does not cover students unless they are regarded as employee trainees. Students, however, are also at risk of workplace TB [58]. High annual TB infection rates (i.e., new infections) of 23/100 person years have been recorded in medical students in Johannesburg using the tuberculin skin test (TST) [59], and 19.3/100 person years in nursing students in Zimbabwe [60]. Community health workers are another category of health worker who may be vulnerable to weak benefits or protections in their employer-employee relationship and thus lack proper social protection [36,61]. 

Outside of statutory protection, there are ethical considerations applicable to secondary prevention and occupational health generally. Two areas of impact are stigma and screening. There is now a large body of literature confirming that stigma looms large in health worker attitudes to TB preventive practices, including unwillingness to self-disclose TB disease or participate in employer provided services where confidentiality is of concern [23,37,38,39,62,63,64,65,66]. The reduction of stigma requires an understanding of its context-specific nature and cultural content, e.g., in South Africa, where tuberculosis has a strong association with HIV infection [62]. However, the need for protection of the privacy of affected workers is in tension with the social need for self-disclosure to serve accurate risk assessment and to protect and educate co-workers. A contrasting strategy used in South Africa by a network of health workers affected by TB has been to publicise their status and share their personal experiences as an educational and a mobilising strategy [50,66]. 

Screening programmes, particularly in low-resource settings, need to pay attention to the ethical implications of medically examining workers who have not presented themselves to healthcare professionals. Such considerations have long been part of medical surveillance in occupational health practice, and a body of ethical and legally sensitive guidelines have been developed [67]. These address issues of informed consent and refusal of consent, the reliability of the test, confidentiality, and third-party reporting of results to employers, all of which require an agreed protocol in advance of screening.

## 6. Heath Worker Voice and Advocacy

Historically in some countries, labour unions have played an important role in securing recognition of the TB risk to health workers and related action [68]. This has included efforts to include a labour perspective in early CDC deliberations on preventive guidelines [69]. However, it is difficult to make a general statement about the current state of labour union involvement globally in the protection of health workers against TB. 

In South Africa, advocacy by health workers themselves has emerged as a prominent voice for health worker rights in the form of TB Proof, a voluntary group that includes a number of members who have suffered from TB [66]. Their work is augmented by a network of concerned health workers internationally [63,64]. TB Proof activities have a several elements that we believe should be emulated elsewhere. These include engagement with national policy-making on TB; maintaining a website with educative and activist materials; a strong focus on destigmatising TB; the targeting of students and junior healthcare staff to protect themselves, but also to assert their right to be protected; and the effective use of personal narratives and media [66].

The voice of health workers has found a place in the large number of publications using qualitative research—typically key informant but also arts-based methods—and covering all aspects of occupational TB risk and management [21,23,25,26,28,35,37,38,39,61]. Some of this work throws light on unrecognised deficiencies in health care practice, such as the exposure of community health workers to infective risk [61] or the failure of patients to accept or understand IPC owing to cultural or language barriers [28]. 

## 7. Conclusions

Guidelines aimed at standardising operational practice to prevent occupational TB are essential. However, implementation takes place in a local setting, characterised by its own legal framework and employee rights regimen, resourcing, co-morbidity (such as HIV), and cultural attitudes. As Greenhalgh and Papoutsi have argued, the complexity lies in the interaction between an intervention and the pre-existing organisation of health care, and not necessarily in the intervention on its own [70]. For example, barriers to implementation can be lowered by intensified training of health workers [71,72], but the argument here is that such training is insufficient for sustainability if the necessary systemic scaffolding is not in place. The experience of two of the authors (A.Y. and J.S.) in an occupational health/IPC programme, developed over almost 10 years in one of South Africa’s poorer provinces, illustrates the value of customized interventions, including new policies and staffing, at the individual, facility, and provincial and national government levels [34,42]. 

A common approach to improving implementation is auditing, using operational checklists as a basis for expected quality improvement [41,73]. However, what we propose here is that the concept of a checklist be expanded to include the widest perspective possible. It should cover questions such as whether the system includes primary, secondary, and tertiary levels of protection, as well as embracing a health system framework such as the ones we have described here; whether there is explicit commitment of senior leadership to health system strengthening via IPC and occupational health and safety; whether the legal and ethical implications in relation to screening, coverage, and other thorny aspects referred to earlier are being dealt with; and whether channels for worker voice and agency exist and are used. While not easy to achieve, particularly in high-TB-burden, low-resource settings, policies and practices that incorporate this approach are more likely to provide for long-term sustainable protection of the essential human resources needed to fight TB and indeed other infectious hazards at work.

### Postscript

This commentary was prepared before the COVID-19 crisis. While there are many differences between tuberculosis and COVID, the approach outlined provided a guide in the early phases of the local COVID epidemic in South Africa. Over and above the urgent pressures of IPC, a systems approach has enabled recognition of the need for collaboration across disciplines and organisational units, occupational health coverage of all levels of the health system, a rapid-response information system on health worker infection and attrition, and a properly functioning worker’s compensation regimen.

As the COVID pandemic develops, it is likely that many of the elements identified in this commentary will require closer attention. What has been striking in the scramble worldwide to put the necessary protective systems in place is the vulnerability of health workers even in high-income countries. However, the longer view must be taken. Robust systems designed to protect health workers from infectious hazards, whether viral epidemics or TB, are needed as a continuing rather than as a reactive project, to better withstand the further threats to come. This includes making use of the crisis to embed properly functioning occupational health and IPC practices as an essential part of a resilient health system.

## Figures and Tables

**Figure 1 ijerph-17-03957-f001:**
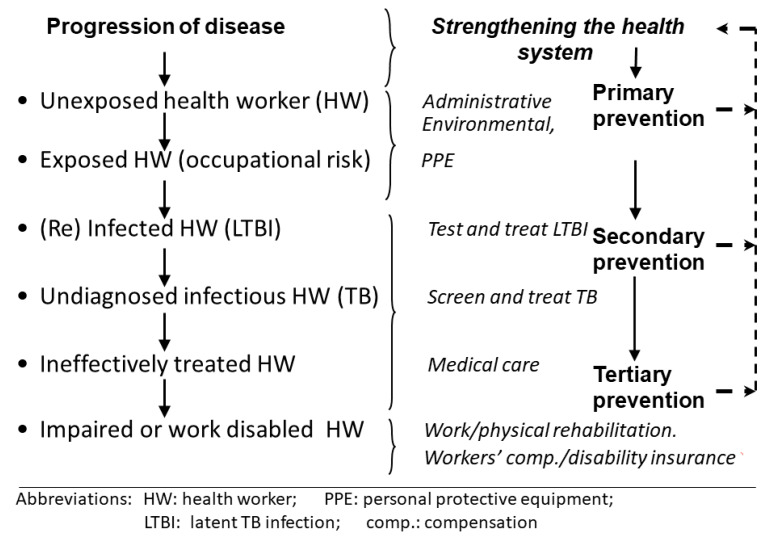
TB disease prevention schema.

**Table 1 ijerph-17-03957-t001:** Components of a comprehensive occupational health approach to the protection of health workers from occupational tuberculosis (TB).

	Domain	Definition
1	Levels of disease prevention	Primary (control of TB transmission in healthcare settings), secondary (TB screening, early diagnosis of health workers, and effective treatment), and tertiary (rehabilitation, jobs, and social security).
2	Health systems framework	Assessing and strengthening the capacity of the health system to deliver quality healthcare to the whole population, including its own workforce, across all medical conditions, and to respond to crises.
3	Law and ethics	Understanding and use of statutory legal instruments, as well as the management of ethical implications of practices to protect health workers from TB.
4	Health worker voice and advocacy	Extent to which opportunities are provided for the views and experiences of health workers to be raised to influence the organisation of healthcare.

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
