# Peer review of "Current Guidelines for Protecting Health Workers from Occupational Tuberculosis Are Necessary, but Not Sufficient: Towards a Comprehensive Occupational Health Approach"

_ijerph, 2020, doi:10.3390/ijerph17113957_

Round 1
Reviewer 1 Report
The scourge of tuberculosis with its special enmity for health workers, long pre-dated our current age of COVID and will be awaiting us when we return our attention to it after COVID is held at bay in the coming months. The authors make a case for the limits of IPC approaches alone and the need for a more comprehensive approach to TB control, and I agree. To hear in the “postscript” that this approach has also proven helpful to manage the COVID onslaught in South Africa is an added endorsement of such a collaborative process. Certainly, the piece makes a convincing and well-referenced argument for this in the case of TB. One reflection I would make is, while you well describe the broad architecture of such a more comprehensive plan, might you suggest some examples for the reader of how such a collaboration would work for some specific element of TB prevention?
For example, you state the need to engage the “scaffolding” of health systems and have leadership commitment to implement even the IPC program elements; here you could then suggest examples of specific programs or initiatives that might be a successful initial effort with such leadership support. Do you have ideas of which element of an IPC protocol should be launched first, as a pilot of the comprehensive collaborative approach? Also, I have wondered if one of the reasons, besides those you mention here, that IPC efforts are not more robust, and I believe one author has even called “nonexistent” in some high TB burdens countries, is the lack of a metric of success (or failure), such as TB skin test surveillance.
However, the piece is also fine as is and I think readers in occupational and public health would appreciate the carefully argued model you propose. Now, if we can convince highly placed thought leaders and funders….
Author Response
Thank you for the comments.
Please see added text (in italics) and a comment below each item.
- One reflection I would make is, while you well describe the broad architecture of such a more comprehensive plan, might you suggest some examples for the reader of how such a collaboration would work for some specific element of TB prevention
Current P. 5: “..This creates hesitancy among senior health service managers to commit significant resources to occupational health and safety amidst competing priorities, even if the actions are legally mandated, and especially if they believe TB in their staff is not an occupational disease [38, 40]. This suggests that a useful starting point for collaboration between occupational health and IPC is an information system able to track and investigate active TB in health workers for purposes of risk assessment and targeted IPC. An example is the Occupational Health and Safety Information System (OHASIS) developed through a collaboration of occupational health and IPC professionals at the University of British Columbia with the National Institute for Occupational Health in South Africa and applied in the National Health Laboratory Service of South Africa [53, 54]. “
(Comment: If you can’t measure you can’t manage it.)
- For example, you state the need to engage the “scaffolding” of health systems and have leadership commitment to implement even the IPC program elements; here you could then suggest examples of specific programs or initiatives that might be a successful initial effort with such leadership support. Do you have ideas of which element of an IPC protocol should be launched first, as a pilot of the comprehensive collaborative approach?
Current p. 8: “..For example, barriers to implementation can be lowered by intensified training of health workers [70,71] but the argument here is that such training is insufficient for sustainability if the necessary systemic scaffolding is not in place. The experience of two of the authors (AY, JS) in an occupational health/IPC programme developed over almost ten years in one of South Africa’s poorer provinces illustrates the value of customized interventions, including new policies and staffing, at individual, facility and provincial and national government level [34, 42]. "
(Comment: Not so much a “pilot” as a sustained campaign to change the way things are done throughout the system, which takes time. Also, it needs to be multi-element rather focused on one element.)
- Also, I have wondered if one of the reasons, besides those you mention here, that IPC efforts are not more robust, and I believe one author has even called “nonexistent” in some high TB burdens countries, is the lack of a metric of success (or failure), such as TB skin test surveillance.
Current p. 4 : “.. However, secondary prevention contributes to primary prevention by aiming to keep health workers unimpaired and non-infectious in their daily work. This includes occupational health services to provide screening for active TB, and early diagnosis and management of affected health workers [16]. An occupational health platform is also needed for screening for and treatment of LTBI. While such screening has long been recommended in high-income countries such as the United States [7, 10], it appears in the latest WHO guideline as a (conditional) recommendation for low TB incidence countries only [43]. Consensus on what is required and feasible in high TB incidence LMIC settings is elusive. While there are many studies of LTBI prevalence in these settings, programme implementation research in this area is scarce and is a pressing need."
(Comment: This response is linked with the response under point 1 on the need for an information system for surveillance.)
Reviewer 2 Report
Dear Authors
I carefully evaluated the paper, finding it overall well written and well presented. The theme is interesting and there is a need to investigate these aspects.
A comprehensive approach could be very useful in occupational health settings, especially in high prevalence TB zones.
About the primary prevention, do you think that the TB vaccination addressed to health care workers when working in high risk wards (e.g. pneumology, infectus disease specialist) in high prevalence countries, could play a significant role on protecting them? In the primary prevention section, I suggest to discuss if a vaccination strategy, when properly addressed, can be considered as a part of a comprehensive approach in occupational health settings.
Best regards
Author Response
Thank you for the comments.
Please see added text (in italics).
- About the primary prevention, do you think that the TB vaccination addressed to health care workers when working in high risk wards (e.g. pneumology, infectious disease specialist) in high prevalence countries, could play a significant role on protecting them? In the primary prevention section, I suggest to discuss if a vaccination strategy, when properly addressed, can be considered as a part of a comprehensive approach in occupational health settings.
Current p .4: “Vaccination of health workers against TB also falls into the category of primary prevention. BCG vaccination or revaccination of health workers is not currently a recommendation by either CDC [10] nor WHO, and variation in health worker BCG vaccination practice across Europe reflects lack of consensus as to its efficacy [45]. However, vaccination should remain on the agenda of a comprehensive occupational health approach, whether, for example, BCG revaccination of LTBI negative health workers or eventual testing and roll out among health workers of one of the new vaccines on the horizon [46,47].”